# Identification of upstream transcription factor binding sites in orthologous genes using mixed Student's *t*-test statistics

**Tinghua Huang, Hong Xiao, Qi Tian, Zhen He, Cheng Yuan, Zezhao Lin, Xuejun Gao**[ID]\*, **Min Yao**[ID]\*

College of Animal Science, Yangtze University, Jingzhou, China

\* gaoxj53901@163.com (XG); minyao@yangtzeu.edu.cn (MY)

**Data Availability Statement:** Grit is publicly available at http://www.thua45.cn/grit under an academic free license.

**Funding:** This project was funded by the National Natural Science Foundation of China NSFC No.

## Abstract

### Background

Transcription factor (TF) regulates the transcription of DNA to messenger RNA by binding to upstream sequence motifs. Identifying the locations of known motifs in whole genomes is computationally intensive.

### Methodology/Principal findings

This study presents a computational tool, named "Grit", for screening TF-binding sites (TFBS) by coordinating transcription factors to their promoter sequences in orthologous genes. This tool employs a newly developed mixed Student's *t*-test statistical method that detects high-scoring binding sites utilizing conservation information among species. The program performs sequence scanning at a rate of 3.2 Mbp/s on a quad-core Amazon server and has been benchmarked by the well-established ChIP-Seq datasets, putting Grit amongst the top-ranked TFBS predictors. It significantly outperforms the well-known transcription factor motif scanning tools, Pscan (4.8%) and FIMO (17.8%), in analyzing well-documented ChIP-Atlas human genome Chip-Seq datasets.

### Significance

Grit is a good alternative to current available motif scanning tools.

### Author summary

Locating transcription factor-binding (TF-binding) site in the genome and identification their function is fundamental in understanding various biological processes. Improve the performance of the prediction tools is important because accurate TF-binding site prediction can save cost and time for wet-lab experiments. Also, genome wide TF-binding site prediction can provide new insights for transcriptome regulation in system biology perspective. This study developed a new TF-binding site prediction tool based on mixed

31902231 (MY) and 31402055 (TH), the College Students' Innovation and Entrepreneurship Training Program of Yangtze University No. 2020084 (TH), the Teaching research project of Yangtze University No. JY2020125 (TH), and the Graduate Teaching Program of Yangtze University No. YAL202108 (MY). The funders had no role in study design, data collection and analysis, decision to publish, or preparation of the manuscript.

**Competing interests:** The authors have declared that no competing interests exist.

Student's *t*-test statistical method. The tool is amongst the top-ranked TF-binding site predictors, as such, it can help the researchers in TF-binding site identification and transcriptional regulation mechanism interpretation of genes.

This is a *PLOS Computational Biology* Software paper.

## 1. Introduction

A DNA sequence motif is a short conserved pattern that can be coordinated by regulator proteins, such as transcription factors (TFs). DNA sequence motifs represent functionally important regions of the genome and are one of the basic units of molecular evolution that are usually conserved among species [1]. Locating these motifs in the genome and understanding their function is fundamental in building molecular models for biological processes such as human diseases [2, 3]. Researchers often face the task of identification of putative binding sites for TFs in whole genomes, termed "motif scanning" [4]. Over the past several decades, many computational pipelines have been described that utilize position weight matrices (PWM) for this task.

MAST searches DNA motifs against a database composed of short sequences and assigns a score to each target sequence assuming that every motif occurs once [5]. MCAST uses a hidden Markov model (HMM) to scan DNA sequences for regions comprising one or more of the given motifs [6], whereas SWAN utilizes a log likelihood ratio (LLR) scoring system built by training a two-state HMM on the background sequences [1]. FIMO computes a LLR score for each position in a DNA sequence motif and converts this score to a *q*-value using dynamic programming methods [7]. TRAP introduces a physical binding model to predict the relative binding affinity of a transcription factor for a given sequence [8]. PWMScan scans sequence motifs using Bowtie [9] or "matrix_scan", which employs a conventional search algorithm [10]. The Python-based program Motif scraper searches motifs specified as a text string using IUPAC degenerate bases [11]. Several tools such as Toucan [12], OTFBS [13], and CREME [14] count all matches in the target and control sequences and apply binomial statistics for over-representation. Other tools such as Clover [15], PAP [16], oPOSSUM [17], Pscan [18], and TFM_Explorer [19] scan sequence sets from co-regulated or co-expressed genes with TF motifs, and assess motifs that are significantly over- or under-represented, to identify common regulators of the sequence sets. WeederH was designed for discovering conserved TFBS and distal regulatory modules in sequences from orthologous genes [20]. MatrixREDUCE predicts functional transcription factor binding through alignment-free and affinity-based analyses of orthologous promoter sequences in closely related species [21]. Table 1 summarizes motif scanning parameters of 18 currently available tools.

To overcome the shortcomings of currently available tools as described below, a novel motif-scanning algorithm "Grit" was developed that identifies genome-wide upstream TFBS from a known collection of PWMs for promoters of orthologous genes without the need of sequence alignment. This study addressed the question of finding significant associations between TFs and orthologous gene sets by introducing a new computational framework that uses mixed Student's *t*-test statistics and adopts new ways of constructing promoter sequence sets of orthologous genes. Its application to the human genome has yielded fruitful results, demonstrating its desirability as a motif scanning tool.

## 2. Design and implementation

### 2.1. Building putative promoter sets for orthologous genes

PWMs for TFs were obtained from the Jaspar database release 2020, referred here as JASPAR-2020 [22], which comprises a collection of TF motifs for human species. The Ensembl Biomart web tool release 100 [23] was used to extract the putative promoter sequences 2 kb upstream of the transcript, for all genes in 294 genomes (S1 Table). The promoter set for orthologous genes used for scanning of TFBS was built by first comparing the cDNA sequence from the target genome (TG, human) with the cDNA sequence from the other 293 reference genomes (RGs, genomes other than human) to identify the orthologous gene clusters, and consequently, put the 2 kb upstream sequence of the orthologous genes together. The BLASTN parameter was "-word_size 11 -reward 2 -penalty -3 -gapopen 5 -gapextend 2 -evalue 1e-6", and the BEST-to-BEST approach based on the E-value (mutual best hit) was used to define orthologous gene pairs between the two species. This promoter set was referred to as the "2K-set" and was available from the Grit website, the promoter sequence for the TG in this set was referred to as the "TPS". A random background promoter sequences set was randomly selected from the 2K-set and named as "BPS".

### 2.2. Statistical identification of TFBS in a target genome

First, we obtained a statistical score based on a component of HMM (Eq 1). The implementation of this raw score (*RS*) represented the ideals of existing statistical approaches [15].

$$RS = ln \frac{1}{M_s} \sum_{L=1}^{M_s} \prod_{k=1}^{w} \frac{q(k, L_k)}{p(L_k)}, 1 \le s \le l - w + 1 \tag{1}$$

**Table 1. Functionality of currently available motif scanning tools.**

| Tool | Scan single DNA | Scan multiple DNAs^a | Report single *p*-value^b | Report multiple *p*-value^c | Species specific | Utilize conservation information | Source code available | Compared^d | Release date |
|---|---|---|---|---|---|---|---|---|---|
| MAST | ✓ | | ✓ | | ✓ | | ✓ | | 1998 |
| MCAST | ✓ | | ✓ | | ✓ | | ✓ | | 2003 |
| OTFBS | | ✓ | | ✓ | | ✓ | | | 2003 |
| CREME | | ✓ | | ✓ | | ✓ | | | 2003 |
| Clover | | ✓ | | ✓ | | ✓ | ✓ | ✓ | 2004 |
| Toucan | | ✓ | | ✓ | | ✓ | | | 2005 |
| PAP | | ✓ | | ✓ | | ✓ | | | 2006 |
| oPOSSUM | | ✓ | | ✓ | | ✓ | | | 2007 |
| TRAP | ✓ | | | | | ✓ | ✓ | | 2007 |
| WeederH | | ✓ | | ✓ | | ✓ | ✓ | | 2007 |
| MatrixREDUCE | | ✓ | | ✓ | | | ✓ | | 2008 |
| Pscan | | ✓ | | ✓ | | ✓ | ✓ | ✓ | 2009 |
| TFM_EXPLORER | | ✓ | | ✓ | | ✓ | ✓ | | 2010 |
| SWAN | ✓ | | ✓ | | | ✓ | ✓ | ✓ | 2010 |
| FIMO | ✓ | | ✓ | | ✓ | | ✓ | ✓ | 2011 |
| PWMScan | ✓ | | ✓ | | | | ✓ | ✓ | 2018 |
| Motif scraper | ✓ | | | | ✓ | | ✓ | | 2018 |
| Grit | | ✓ | ✓ | | ✓ | ✓ | ✓ | ✓ | 2021 |

^a Designed to scan multiple sequence sets.

^b Report *p*-value for target genome.

^c Report *p*-value for multiple sequence sets.

^d Selected for performance assessment.

The *RS* calculation represents repeated averaging of LLRs. *RS* represented the LLR for a motif being present at one particular location in a sequence, where *w* was the width of the motif, *L* denoted the location being considered, $L_k$ was the nucleotide at position *k* within this location, $p(L_k)$ is the background probability of observing nucleotide $L_k$ estimated from the frequency of $L_k$ in that sequence, and $q(k, L_k)$ is the probability of observing nucleotide $L_k$ estimated from the frequency of the $K_{th}$ location in the motif. The *RS* for a motif present in a sequence with length *l* was the natural logarithm of the average of LRs taken over all locations of *s*, where $M_s$ was the number of locations in the sequence calculated as *l–w* + 1. Statistically significant TFBS in the target genome are identified by a mixed Student's *t*-test.

## 2.3. Theory of mixed Student's *t*-test

We tested the significance of *RS* of a gene in the TG for a given motif assuming that the *RSs* for the sequences in the 2K-set for this gene were normally distributed. We propose a new statistical approach that is an extension of the Student's *t*-test, namely, the "mixed Student's *t*-test". A slightly varied statistical approach from the canonical Student's *t*-test was proposed—giving a background set (*bkg*) and a testing set (*obs*), we determine whether one value (*one*) from the *obs* is significantly different from the mean of the values in *bkg*, where *one*, *obs* and *bkg* are *RSs* for TPS, 2K-set, and BPS, respectively. The mixed Student's *t*-test statistic can be calculated by combining the one-sample Student's *t*-test and independent two-sample Student's *t*-test. The calculation of the *t*-statistics (*t'*) and degree of freedom (*df*) of the mixed Student's *t*-test were presented as Eqs 2 and 3, respectively. The *p*-values can be estimated by the "*cdflib*" function [24].

$$t' = \frac{\bar{X}_{bkg} - one}{\sqrt{\frac{(n_{obs}-1)s_{obs}^2 + (n_{bkg}-1)s_{bkg}^2}{n_{obs}+n_{bkg}-2}} \cdot \sqrt{\frac{1}{n_{obs}} + \frac{1}{n_{bkg}}}} \tag{2}$$

$$df = \frac{\left(\frac{s_{obs}^2}{n_{obs}} + \frac{s_{bkg}^2}{n_{bkg}}\right)^2}{\frac{(s_{obs}^2/n_{obs})^2}{n_{obs}-1} + \frac{(s_{bkg}^2/n_{bkg})^2}{n_{bkg}-1}} \tag{3}$$

The coefficient of conserved variation (*CCV*, Eq 4) and standard difference (*SD*, Eq 5) are calculated, which indicate the degree of conservation of the TFBS among species and the altitude of difference in *RS* scores between the TG and the RGs, respectively.

$$CCV = \frac{|\bar{X}_{obs}|}{\sqrt{\sum_{i=1}^{n_{obs}} (obs_i - one)^2 / n_{obs}}} \tag{4}$$

$$SD = \frac{one - \bar{X}_{bkg}}{|\bar{X}_{bkg}|} \tag{5}$$

## 2.4. Development of the Grit software

Utilizing the mixed Student's *t*-test statistics, we developed a tool, called Grit, for screening TFBS by coordinating TFs to their promoter sequences in orthologous genes. The tool takes JASPAR-2020 (specified by the -m option), 2K-set (-i option), and BPS (-b option) as it's input. Running the tool with default options (-n 10 -z 200 -s 1 -t 0.05 -p 0) will produce a result file (-o option) containing the predicted TFBS. There are three major steps built into the

program: 1) calculate the *RS* for each PWMs presented in each promoter set for orthologous genes using Eqs 1 and 2) calculate the *p*-values for the significance of *RS* of each genes for each given PWMs using the mixed Student's *t*-test statistics; and 3) perform multiple testing correction for all *p*-values using the FDR method [25], and retain the TFBS with FDR $\leq$ threshold defined by the -t option. The source code has been deposited in GitHub and is available under academic free license.

## 2.5. Performance assessment methods of the Grit software

The ReMap datasets were obtained from the ReMap website release 2020, referred to as ReMap-2020 [26], and ChIP-Atlas website release 2021, referred to as Atlas-2021 [27]. True positives (TP) were defined as predicted binding sites overlapping 80% with experiment-supported binding sites from ReMap or the ChIP-Atlas ChIP-Seq datasets. False positives (FP) were defined as predicted binding sites that did not overlap with experiment-supported binding sites, and false negatives (FN) were defined as experiment-supported binding sites that were not identified. Performance was assessed by calculating sensitivity [Sn = TP/(TP + FN)], positive predictive value [PPV = TP/(TP + FP)], and geometric accuracy [ACCg = $\sqrt{Sn \cdot PPV}$], as reported by Sand et al. [28] and Jayaram et al. [29] for all of the datasets analyzed. All assessments of the six tools were performed on Amazon EC2 computation services in parallel. For software such as PWMScan and Clover, where the local version was not available or too slow to analyze all PWMs, a random subset of the transcription factors (35 TFs) was analyzed. The Sn, PPV, and ACCg values for each tool were compared by paired Student's *t*-test.

## 3. Results

### 3.1. Mixed student's t-test with simulated datasets

Two normally distributed datasets were generated and used as bkg (mean = −10, SD = 5, gray) and obs (mean = −2, SD = 7, dark green) datasets. Three values from the obs (located at 25, 50, and 75 percentiles, red) were tested for their significance, and produced p-values of 1.0, 0.03, and 1E-25, respectively (Fig 1A). Three representative genes from the human genome were used for testing: purple for a gene at the 75% quantile of *CCV* and 25% quantile of *SD*, dark green for a gene at the 50% quantile of *CCV* and 75% quantile of *SD*, and red for a gene at the 25% quantile of *CCV* and 75% quantile of *SD*, all of which produced *p*-values less than 1E-6 (Fig 1B). The *p*-values, *CCV*, and *SD* for each entity (*one*) in *obs* were calculated. As shown in Fig 1C, the *p*-values decreased with increasing *CCV* and *SD*, indicated that the mixed Student's *t*-test prefers TFBS either having higher *CCV* or having higher *SD*, or both. We also noted that the mixed *t*-test behaved as a one-sample *t*-test when distributions of values in *obs* and *bkg* were the same, or a two-sample *t*-test when observation (*one*) was located at the mean of the *obs* (S1 Text).

### 3.2. Prediction of TFBS in human genome using Grit

A schematic of the pipeline used in this study was indicated in Fig 1D, which included: (1) blast TG with RGs; (2) build the 2K-set for homolog genes using the BEST-to-BEST approach; (3) run Grit using Jaspar-2020 and the 2K-set; and 4) assess the performance of Grit using public ChIP-Seq datasets. The promoter set contained 2 kb length sequences for a putative promoter region of 35,342 homologues gene clusters. To estimate the accuracy of this promoter set, we compared it with experimentally validated human promoters available in the EPD database containing 29,598 human gene promoter sequences [30]. The TPS contained promoter sequences for 93.2% of these genes showing post alignment with the TPS of the EPD sequences with an E-value $< 1E^{-6}$.

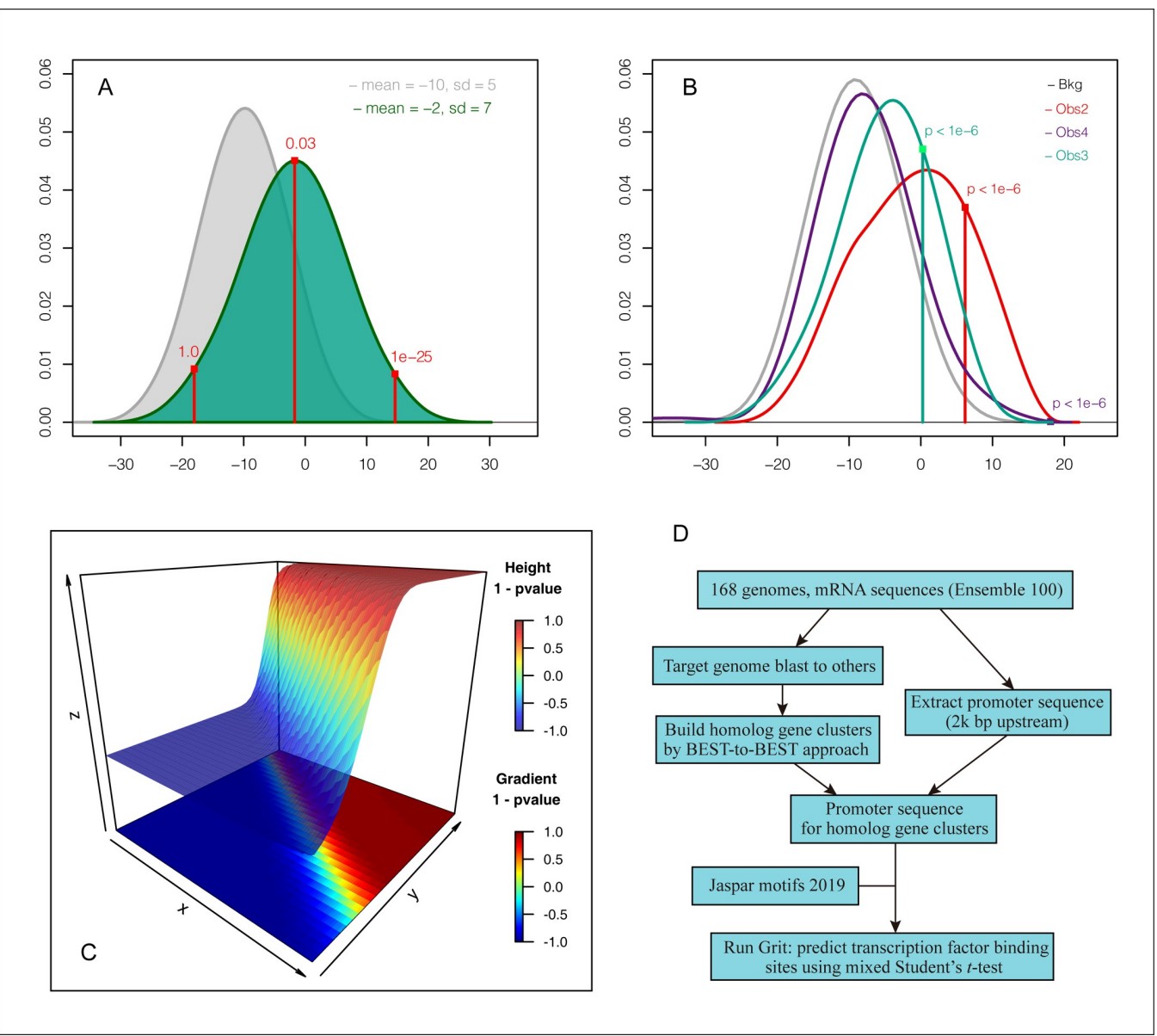

**Fig 1. Validating the mixed Student's *t*-test with simulated and real-world data.** A and B. Simulated data and representative real-world data testing; the distribution of *bkg* was colored with gray, and *obs* with dark green; the *one* values of interest were indicated with vertical lines. C. Parameter testing. The X axis represented the coefficient of conserved variation (*CCV*, 0 to 2.0), the Y axis represented the standard deviation (*SD*, 0 to 3.0), and the Z axis represented the 1 – *p*-value. D. Schematic of the pipeline used for the study.

Grit was used for identification of TFBS in the human genome by applying it to the 2K-set datasets. The Grit run took 22 h and identified 7.57 million significant TFBS for 537 TFs (FDR ≤ 0.05). A target gene was assigned a TF if the gene was found in at least one TFBS. Grit prediction results were assessed with six publicly available motif scanning tools designed for high throughput analysis using the 829 ReMap-2020 datasets (S2 Table) obtained from the ReMap database [26]. The results were shown in Fig 2. FIMO and Swan consistently achieved higher Sn but lower PPV for ChIP-Seq datasets as compared with other tools (*p*-value ≤ 0.05). The average Sn of Grit is lower than FIMO but the average PPV of Grit is the highest among all competitors. As results, Grit attained the highest average ACCg, followed by FIMO, Swan,

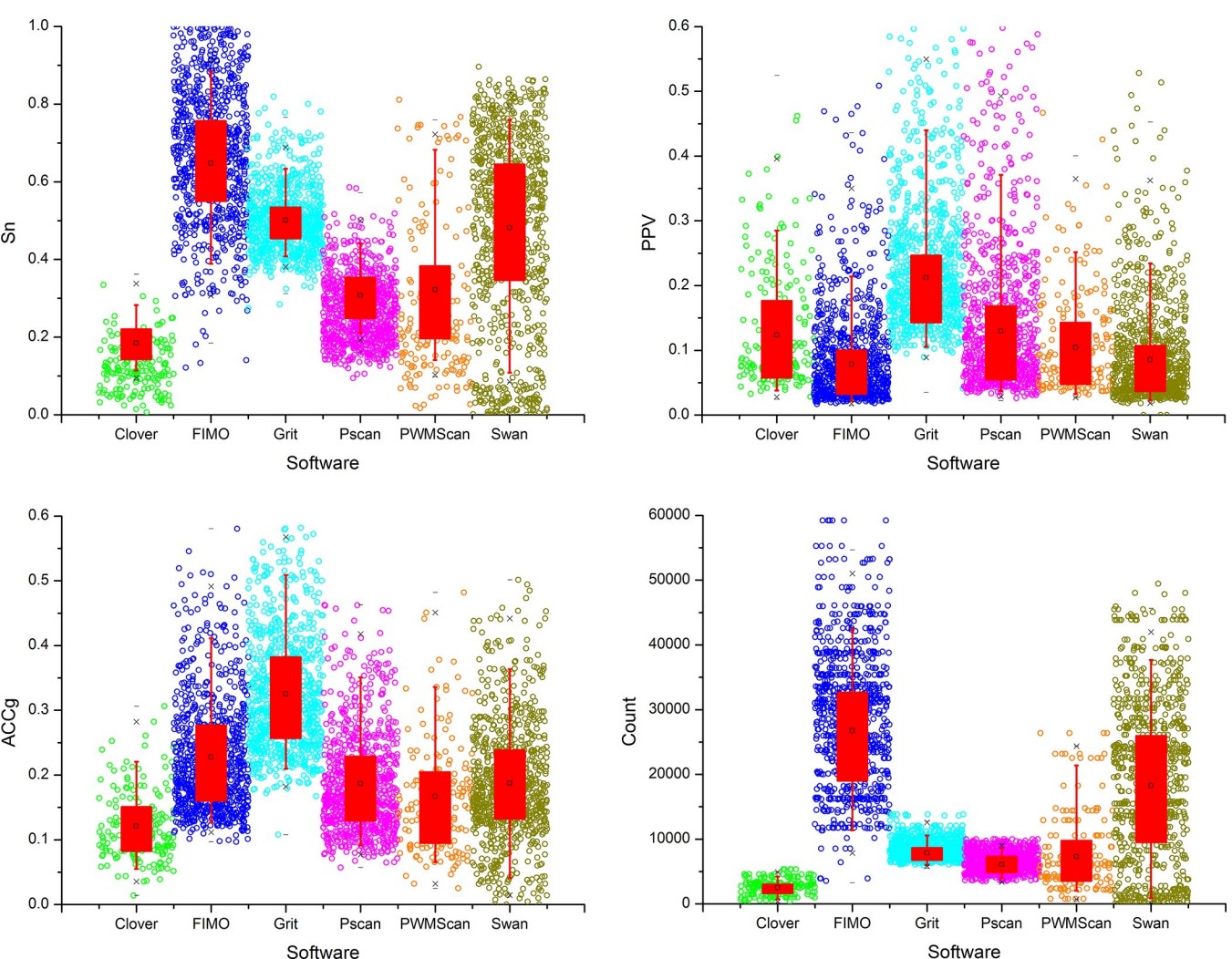

**Fig 2. Performance assessment of motif scanners in analyzing ReMap datasets.** A total of six scanners (Clover, FIMO, Grit, Pscan, PWMScan, and Swan) were evaluated based on the parameters of sensitivity (Sn), positive predictive value (PPV), geometric accuracy (ACCg), and total number of predicted transcription factor binding sites (TFBS, Count).

Pscan, and PWMScan, Clover had the lowest ACCg. It is noticed that Grit outperformed FIMO 29% as evaluated by ACCg (*p*-value ≤ 0.05). The number of predicted targets for FIMO and Swan was strikingly high, covering approximately 80% of human genes on average, whereas the number of predicted targets for Grit, Pscan, and PWMScan were significantly smaller, with Clover producing the lowest targets (*p*-value ≤ 0.05).

### 3.3. Performance of Grit using ChIP-Atlas datasets

Additionally, performance of the six scanners was evaluated using 111 high-quality Atlas-2021 target gene sets (S3 Table) collected from experimentally validated human ChIP-Atlas data with literature support [27]. Table 2 listed a random subset of the assessment results. The Sn values of FIMO were higher than those of Grit (33.0%, *p*-value ≤ 0.05), whereas the PPV values for Grit were higher than those of FIMO (2.15 fold, *p*-value ≤ 0.05). The ACCg values for Grit were higher than those of FIMO (17.8% on average, *p*-value ≤ 0.05), indicating that Grit performed better than FIMO for Atlas-2021. Furthermore, the Grit method slightly outperformed

**Table 2. Performances of Grit with FIMO using publicly available Chip-seq datasets with literature support.**

| Motif | Grit | | | FIMO | | | Pscan | | |
|---|---|---|---|---|---|---|---|---|---|
| | Sn | PPV | ACCg | Sn | PPV | ACCg | Sn | PPV | ACCg |
| ASCL1 | 0.51 | 0.15 | 0.28 | 0.94 | 0.04 | 0.20 | 0.46 | 0.16 | 0.27 |
| CDX2 | 0.44 | 0.18 | 0.28 | 0.69 | 0.07 | 0.22 | 0.43 | 0.20 | 0.30 |
| DUX4 | 0.55 | 0.08 | 0.21 | 0.78 | 0.04 | 0.18 | 0.53 | 0.09 | 0.22 |
| E2F1 | 0.61 | 0.80 | 0.70 | 0.31 | 0.49 | 0.39 | 0.58 | 0.81 | 0.69 |
| ELK4 | 0.69 | 0.22 | 0.39 | 0.93 | 0.10 | 0.31 | 0.71 | 0.35 | 0.50 |
| FLI1 | 0.48 | 0.77 | 0.61 | 0.56 | 0.44 | 0.49 | 0.42 | 0.81 | 0.58 |
| GATA3 | 0.48 | 0.68 | 0.57 | 0.50 | 0.32 | 0.40 | 0.52 | 0.67 | 0.59 |
| GLI2 | 0.52 | 0.37 | 0.44 | 0.70 | 0.17 | 0.35 | 0.30 | 0.33 | 0.31 |
| HNF4G | 0.46 | 0.27 | 0.35 | 0.86 | 0.10 | 0.30 | 0.29 | 0.29 | 0.29 |
| JUND | 0.35 | 0.71 | 0.50 | 0.57 | 0.33 | 0.43 | 0.30 | 0.69 | 0.46 |
| MAFF | 0.44 | 0.18 | 0.28 | 0.67 | 0.08 | 0.23 | 0.49 | 0.20 | 0.31 |
| MEF2A | 0.57 | 0.28 | 0.40 | 0.87 | 0.10 | 0.29 | 0.54 | 0.29 | 0.39 |
| MXI1 | 0.40 | 0.72 | 0.54 | 0.59 | 0.38 | 0.47 | 0.34 | 0.72 | 0.49 |
| NFE2 | 0.50 | 0.62 | 0.56 | 0.62 | 0.29 | 0.42 | 0.27 | 0.56 | 0.39 |
| NFIC | 0.34 | 0.59 | 0.45 | 0.73 | 0.26 | 0.44 | 0.32 | 0.58 | 0.43 |
| NR2C2 | 0.49 | 0.13 | 0.25 | 0.91 | 0.05 | 0.21 | 0.31 | 0.12 | 0.19 |
| NRF1 | 0.71 | 0.72 | 0.71 | 0.87 | 0.38 | 0.58 | 0.49 | 0.78 | 0.62 |
| OTX2 | 0.60 | 0.51 | 0.55 | 0.47 | 0.26 | 0.35 | 0.37 | 0.56 | 0.46 |
| PAX5 | 0.59 | 0.63 | 0.61 | 0.82 | 0.29 | 0.49 | 0.50 | 0.63 | 0.56 |
| RUNX3 | 0.37 | 0.60 | 0.47 | 0.56 | 0.28 | 0.40 | 0.26 | 0.65 | 0.41 |
| SP1 | 0.75 | 0.81 | 0.78 | 0.98 | 0.27 | 0.52 | 0.50 | 0.83 | 0.65 |
| SPI1 | 0.42 | 0.78 | 0.57 | 0.78 | 0.31 | 0.49 | 0.46 | 0.78 | 0.60 |
| SRF | 0.36 | 0.63 | 0.48 | 0.41 | 0.31 | 0.36 | 0.46 | 0.62 | 0.53 |
| TBX21 | 0.44 | 0.59 | 0.51 | 0.74 | 0.26 | 0.44 | 0.18 | 0.56 | 0.32 |
| TCF7L2 | 0.40 | 0.62 | 0.50 | 0.71 | 0.25 | 0.42 | 0.29 | 0.59 | 0.42 |
| Average | 0.50 | 0.51 | 0.48 | 0.70 | 0.24 | 0.37 | 0.41 | 0.51 | 0.44 |

*Full information available in S3 Table. Sn = sensitivity, PPV = positive predictive value, ACCg = geometric accuracy.

the Pscan method (4.8% on average, *p*-value ≤ 0.05). Analysis using JASPAR-2020, ReMap-2020, and Atlas-2021 datasets identified Grit, Pscan, and FIMO as the best tools for identifying TFBS (complete prediction results for all the tools have been provided on the Grit website), ranking them based on ACCg in the order Grit > Pscan > FIMO > Swan > PWMScan > Clover.

## 3.4. Differences among Grit and other prediction tools

The prediction results of Grit and five other tools were compared. There were 38.9% TFBS in ChIP-Atlas datasets that were not identified by the other five prediction tools; 32.8% of TFBS were identified by both Grit and the other tools; 11.5% of TFBS were identified by the other tools but not by Grit; and 16.8% of TFBS were identified by Grit but not by the other tools. A total of 2.9% best TFBS identified by Grit for the same gene did not overlap with those identified by the other tools. A comparison of the numbers of TFBS identified by Grit and by the other five tools showed that each tool produced dramatically different prediction results (Fig 3A and 3B). To show the unique features between Grit and the other tools, we investigated the distributions of *CCV* and *SD* for Grit TFBS and Grit specific TFBS (TFBS detected by Grit but did not by other tools, Grit–other, the "–" symbol means subtracting). The results indicated

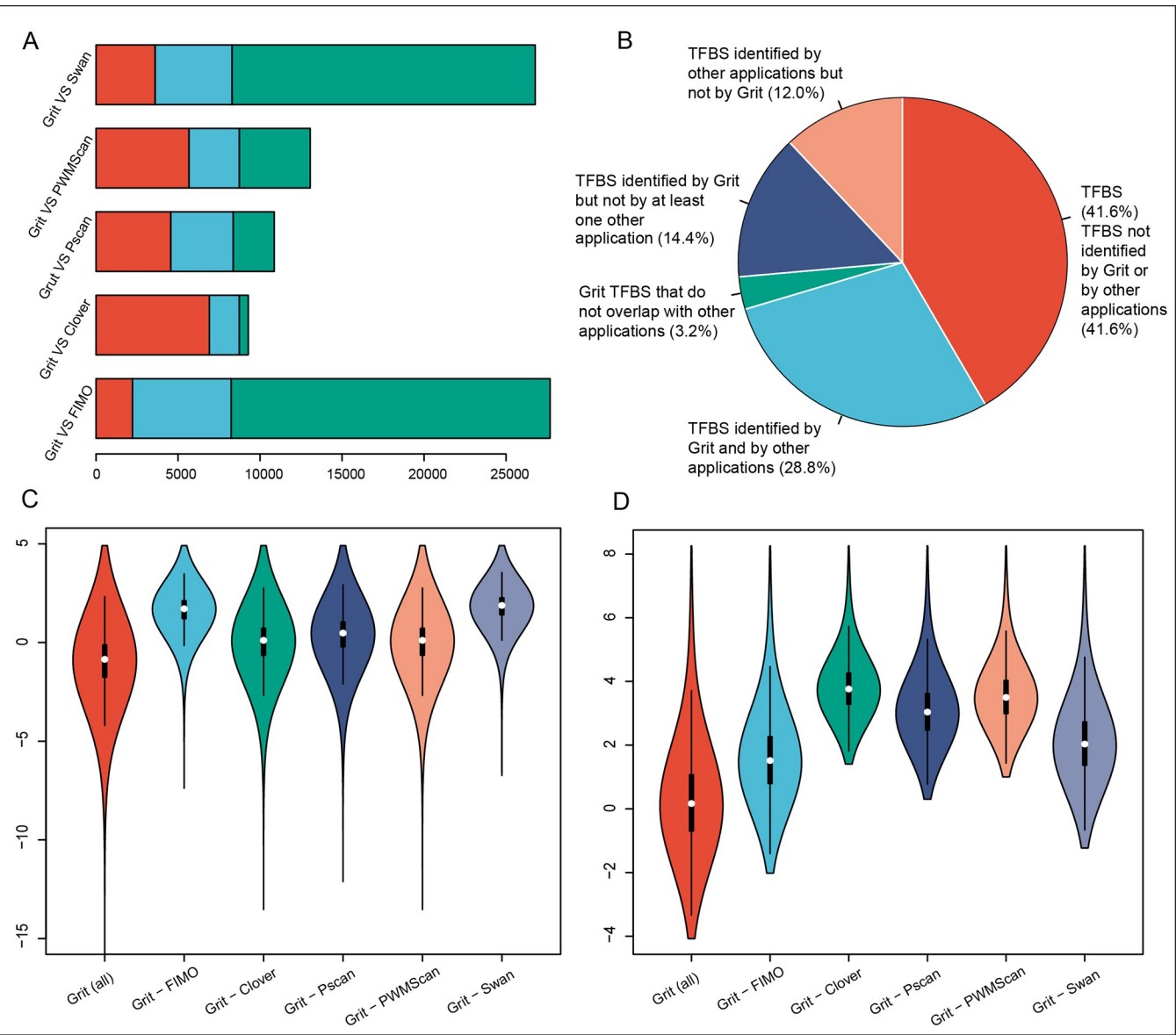

**Fig 3. Differences among the results of Grit and other prediction tools.** A. Number of transcription factor binding sites (TFBS) that overlap (light green) between Grit and other prediction tools, and Grit specific (red) or other tool specific (light blue) TFBS. B. Overall comparison of numbers of TFBS between prediction results of Grit and other prediction tools. The violin plot in C and D shown distributions of conserved variation (*CCV*) and standard difference (*SD*) for TFBS identified by Grit but did not by other tools (Grit–other, the symbol "−" means subtracting), respectively.

that Grit − FIMO and Grit − Swan had significant higher *CCV* values, while Grit − Clover, Grit − Pscan, and Grit − PWMScan had significantly higher *SD* values, than Grit TFBS (p-value $\leq$ 0.05, Fig 3C and 3D).

## 4. Discussion

### 4.1. Comparative genomics is required for TFBS prediction

Identification of TFBS is essential for understanding how TFs regulate gene expression, ultimately controlling processes such as cell cycle progression, stress response, or stem cell

differentiation into adult tissues [31–33]. A typical computational issue is deciding, giving a PWM, if a nucleotide sequence contains a valid instance of the TFBS modeled by the PWM itself [4]. Reliable predictions on a single sequence are nearly impossible without further filtering because of the redundant information available on promoter sequences [18]. The activities of functionally important TFs are highly conserved among both closely related and distant species, thereby causing frequent occurrence of their binding sites in orthologous genes [1]. A gene can be compared with its orthologs by analyzing sequence conservation in evolutionarily preserved transcribed regions, which enables the identification of orthologous gene sets, and TFBS can be predicted from the promoter sequences of these genes [21]. Although the predicted TFBS require further wet-lab experiment validation, with the increasing availability of PWMs, this *in silicon* approach has gained wide popularity [29]. These functionally important binding sites in closely related species can be identified by promoter sequence alignment and phylogenetic printing methods [16, 34–36]. However, promoters of orthologous genes in distantly related species are always poorly conserved, and identification of TF binding sites in these sequences is difficult [1, 20]. This study used cross-species comparison to build co-regulated orthologous gene sets, without the need for non-coding sequence alignment. Therefore, this approach is well suited for comparative genomics across large evolutionary divergences, when existing alignment-based methods are not feasible. The rationale is that the promoters of most of the genes targeted by the same TF(s) should contain significantly higher scores for TFBS than some suitably computed numbers obtained from a collection of unrelated genes or a random background model.

## 4.2. Mixed Student's *t*-test is useful in discovering TFBS

By counting the number of matches and mismatches in target and control sequences, over-represented motif analysis was performed using hypergeometric distribution [14, 37]. A more intricate procedure, accounting for sequences with zero, one, two, or three or more matches in the target and control sets has been reported [31]. Several studies have suggested counting all matches in the target and control sequences, and proposed two different binomial formulas for assessing motif over-representation [12, 14]. Notably, the widely used Pscan program calculates an *RS* similar to Clover's z-test to analyze over- or under-representation of TFBS. The *p*-value is computed by counting the number of times a random dataset yields a score higher than the input sequence set. Our tool "Grit" calculates an *RS* similar to Clover and Pscan, *RS* is the average exponent of the standard motif matrix score and is proportional to the factor's total equilibrium occupancy of the TFBS in sequence in a simple thermodynamic model [38–40]. Note that *RS* is a function of the length of the promoter sequence *S*, and if *S* is extended to include nucleotides that do not coordinate the motif, *RS* would decrease. Given sets of equal-length target and control sequences, it is possible to test for significance by ranking the *RSs* from both sets and performing statistical analyses.

The newly developed mixed Student's t-test was performed for TG and RGs for sites where the TFBS were expected to be conserved. Additionally, we considered the possibility of motif variation among species with highly diverged in RGs, such as pig or cattle, because of significant changes in the binding scores of TF among orthologous genes in the 2K-set of reference species. However, in cases of sufficiently large numbers of RGs, the binding affinity scores should show a normal distribution. The statistical analysis prefers to detect TFBS either conserved among species (high *CCV*) or having significant *RS* differences between the target and control sequences (high *SD*), or both. In contrast to the statistical test implemented in other tools, which produce a "whole" *p*-value for the gene set but fail to tell whether a specific sequence has certain TFBS or not, the mixed Student's *t*-test is not only able to utilize the

information from comparative genomics, but also produces a theoretical *p*-value for an individual sequence of interest.

### 4.3. Single- and multi-species prediction tools

FIMO, Swan, and PWMScan were designed to not only identify potential matches to a motif, but also for potential matches that are greater than expected by chance, considering the genomic background [1, 7, 10]. All were designed for TFBS prediction in a single-species and produced a large number of TFBS as expected. Compared with these tools, Grit identified significantly smaller numbers of binding sites, which highlights the major differences between these tools. Grit has been designed to predict TFBS based on PWMs, and these sites were either highly conserved or had high *RS* among the promoter sequences. With the added condition that the TFBS were required to be highly conserved among species, which was not a criterion for single-species scanners, the final lists produced were relatively small, have a higher *CCV*, and were thus likely to be more suitable for further experimental validation.

Clover and Pscan were designed for multi-species TFBS scanning [15, 18]. Similar to the Clover algorithm, Grit computed an *RS* for each input sequence, representing the average likelihood of each TFBS to a promoter. Regulatory regions of higher eukaryotes often contain multiple binding sites for the same transcription factor, with weaker "shadow" copies of the motif also present [41]. Therefore, considering the average score of multiple matches per sequence will likely aid in the discovery of functional motifs. Another issue is the definition of a "background" suitable for assessing the significance of the results obtained. In Clover, this is performed by shuffling the columns of the motif, or by building random sequence sets of the same size and length of the sequence set investigated [15]. However, the algorithm implemented in Clover is computationally intensive, taking 15 days to process 25 PWMs for the human genome. Similar to Pscan, Grit treats the input sequences as a sample taken from a "universe" composed of all promoter sequences available for the species investigated, several subsamples are taken from the universe, with a default size = 200 and n = 10, and used as the background. For each promoter set of orthologous genes, the *RSs* obtained from the input sequence set can be compared with the *RSs* on the subsets randomly taken from the whole genome promoter set, and the *p*-value can be produced by the mixed Student's *t*-test.

### Availability and future directions

Grit is a good alternative to current available motif scanning tools and is publicly available at http://www.thua45.cn/grit under an academic free license. Further directions will be development of algorithms like gene-set enrichment analysis, to analyze transcriptome data.

### Supporting information

**S1 Text. Proof of the mixed Students' *t*-test.**
(DOCX)

**S1 Table. Detailed information for genomes in Ensembl Biomart web tool release 100.**
(DOCX)

**S2 Table. Detailed information of automated annotated ChIP-Seq datasets obtained from ReMap-2020 database.**
(DOCX)

**S3 Table. Detailed information of the publicly available Chip-Seq datasets (Altas-2021) with literature support.**
(DOCX)

## Acknowledgments

We thank Min Yang, Jinhui Liu, Kaihui Dong, Mingjiang Xu, Zhi Chen, Shijia Zhu, Caiyun Jiang, Yongxia Li, Chenglong Li, Liang He, and Shan Jiang for preparing the promoter sequences, ChIP-Atlas dataset, and ReMap dataset.

## Author Contributions

**Conceptualization:** Tinghua Huang.

**Data curation:** Hong Xiao, Qi Tian, Zhen He, Cheng Yuan, Zezhao Lin.

**Formal analysis:** Tinghua Huang.

**Funding acquisition:** Tinghua Huang, Min Yao.

**Investigation:** Tinghua Huang.

**Methodology:** Tinghua Huang.

**Project administration:** Tinghua Huang, Xuejun Gao, Min Yao.

**Software:** Tinghua Huang.

**Writing – original draft:** Tinghua Huang.

**Writing – review & editing:** Xuejun Gao, Min Yao.

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
