## [Decision Letter · Decision Letter 0]

10 Mar 2022

Dear Dr. Yao,

Thank you very much for submitting your manuscript "Identification of upstream transcription factor binding sites in orthologous genes using mixed Student’s t-test statistics" for consideration at PLOS Computational Biology.

As with all papers reviewed by the journal, your manuscript was reviewed by members of the editorial board and by several independent reviewers. In light of the reviews (below this email), we would like to invite the resubmission of a significantly-revised version that takes into account the reviewers' comments.

We cannot make any decision about publication until we have seen the revised manuscript and your response to the reviewers' comments. Your revised manuscript is also likely to be sent to reviewers for further evaluation.

Sincerely,

Manja Marz

Software Editor

PLOS Computational Biology

Manja Marz

Software Editor

PLOS Computational Biology

Reviewer's Responses to Questions

**Comments to the Authors:**

Reviewer #1: Some minor picks:

Line 12: "3.2 Mb/s" - is it mega-base-pairs (Mbp/s)? Please indicate so as to avoid mix up with MB (file size).

Line 44: Suggest change "as mentioned below" to "as described below".

Line 54: Suggest change "referred to here as" to "referred here as", deleting "to".

The download source code package is missing an essential README file.

Some major concerns:

(1)

This manuscript is well written in general, however, it lack a clear description as how Grit is designed to work: what's the input, what's output, what's the process in which what algorithms was used to perform the motif recognitions calculate and how the reference (Jasper database) was used to make decisions, etc. Figure 1 D looks superfacial at this point. It seems necessary to add a section to serve this purpose.

(2)

The approaches in this report appear to be innovative in terms of using a new mixed Student t-test to determine the cut off thresholds. The authors compared the outcomes with this tool against a number of other computational tools. However, is it reasonable to suggest that where discrepancies exist between this and other tools, would the TFBS under the spotlights be further verified (e.g. with wet lab methods) in order to make a stronger claim? At the lease this should be a discussion point, I think.

Reviewer #2: In the present manuscript, Huang et al. describe a computational tool ("Grit") that identifies transcription factor binding sites (TFBS) in (alignment-free) upstream DNA sequences of orthologous genes, based on PWMs (positional weight matrices) and a new mixed Student's t-test. The authors tested their tool on simulated data and Chip-Seq datasets from the human genome ChIP-Atlas and compared it with several state-of-the-art TFBS prediction tools. The study is well-written and, in general well-executed, but comes short at some points:

1. It is not entirely clear what the motivation/idea of the new statistical test is. It seems to work on the used simulated and real datasets, but the authors could elaborate more on the theory and model of their mixed Student's t-test.

2. In Table 1, the authors state that their tool Grit is species-specific and can identify TFBS conserved among species. This statement is quite ambiguous, and since they only showed results on human datasets, it seems that Grit is human-specific, or at least not tested on other species.

3. It is not clear what kind of multiple testing approach the authors used (if at all) to correct their calculated p-Values. This is crucial for their statistics and predictions and cannot be omitted!

4. Which kind of helpful information does Figure 1C provide?

5. In Figure 3B: What is the difference between "TFBS identified by Grit but not other applications" and "Grit TFBS that do not overlap with other applications"?

6. There are several minor typos throughout the manuscript, which should be carefully checked and corrected.

The authors present an interesting variation of PWMs-based TFBS prediction tools, with an, in general, interesting application of a new statistical model. However, as described in the authors' own words, their tool only "marginally outperforms" current state-of-the-art tools based on one human dataset. To improve their study, the authors should emphasize the details and ideas of their statistical model. They should definitely consider implementing multiple testing corrections and finally apply and test their tool on more datasets.

**Have the authors made all data and (if applicable) computational code underlying the findings in their manuscript fully available?**

Reviewer #1: Yes

Reviewer #2: Yes

PLOS authors have the option to publish the peer review history of their article (what does this mean?). If published, this will include your full peer review and any attached files.

Reviewer #1: **Yes: **Zhiliang Hu

Reviewer #2: No
---

## [Decision Letter · Decision Letter 1]

30 Apr 2022

Dear Dr. Yao,

We are pleased to inform you that your manuscript 'Identification of upstream transcription factor binding sites in orthologous genes using mixed Student’s t-test statistics' has been provisionally accepted for publication in PLOS Computational Biology.

Best regards,

Manja Marz

Software Editor

PLOS Computational Biology

Manja Marz

Software Editor

PLOS Computational Biology

Reviewer's Responses to Questions

**Comments to the Authors:**

Reviewer #1: Thank you for your efforts to address my concerns.

Reviewer #2: The authors answered all my questions and concerns about their manuscript to my satisfaction and improved their manuscript accordingly.

**Have the authors made all data and (if applicable) computational code underlying the findings in their manuscript fully available?**

Reviewer #1: Yes

Reviewer #2: Yes

PLOS authors have the option to publish the peer review history of their article (what does this mean?). If published, this will include your full peer review and any attached files.

Reviewer #1: No

Reviewer #2: No

---

## [Editor Report · Acceptance letter]

2 Jun 2022

PCOMPBIOL-D-21-02278R1 

Identification of upstream transcription factor binding sites in orthologous genes using mixed Student’s t-test statistics

Dear Dr Yao,

I am pleased to inform you that your manuscript has been formally accepted for publication in PLOS Computational Biology. Your manuscript is now with our production department and you will be notified of the publication date in due course.

With kind regards,

Agnes Pap
